# Using the Microbiome as a Regenerative Medicine Strategy for Autoimmune Diseases

**DOI:** 10.3390/biomedicines11061582

**Published:** 2023-05-30

**Authors:** Kaitlin L. Williams, Ryan Enslow, Shreyas Suresh, Camille Beaton, Mitchell Hodge, Amanda E. Brooks

**Affiliations:** 1College of Osteopathic Medicine, Rocky Vista University, Parker, CO 80112, USA; 2College of Osteopathic Medicine, Rocky Vista University, Ivins, UT 84738, USA; 3Department of Research and Scholarly Activity, Rocky Vista University, Ivins, UT 84738, USA; 4Department of Research and Scholarly Activity, Rocky Vista University, Parker, CO 80112, USA

**Keywords:** autoimmune, gut dysbiosis, microbiome, regenerative medicine, bone metabolism, arthropathy

## Abstract

Autoimmune (AI) diseases, which present in a multitude of systemic manifestations, have been connected to many underlying factors. These factors include the environment, genetics, individual microbiomes, and diet. An individual’s gut microbiota is an integral aspect of human functioning, as it is intimately integrated into the metabolic, mechanical, immunological, and neurologic pathways of the body. The microbiota dynamically changes throughout our lifetimes and is individually unique. While the gut microbiome is ever-adaptive, gut dysbiosis can exert a significant influence on physical and mental health. Gut dysbiosis is a common factor in various AI, and diets with elevated fat and sugar content have been linked to gut microbiome alterations, contributing to increased systemic inflammation. Additionally, multiple AI’s have increased levels of certain inflammatory markers such as TNF-a, IL-6, and IL-17 that have been shown to contribute to arthropathy and are also linked to increased levels of gut dysbiosis. While chronic inflammation has been shown to affect many physiologic systems, this review explores the connection between gut microbiota, bone metabolism, and the skeletal and joint destruction associated with various AI, including psoriatic arthritis, systemic lupus erythematosus, irritable bowel disease, and rheumatoid arthritis. This review aims to define the mechanisms of microbiome crosstalk between the cells of bone and cartilage, as well as to investigate the potential bidirectional connections between AI, bony and cartilaginous tissue, and the gut microbiome. By doing this, the review also introduces the concept of altering an individual’s specific gut microbiota as a form of regenerative medicine and potential tailored therapy for joint destruction seen in AI. We hope to show multiple, specific ways to target the microbiome through diet changes, rebalancing microbial diversity, or decreasing specific microbes associated with increased gut permeability, leading to reduced systemic inflammation contributing to joint pathology. Additionally, we plan to show that diet alterations can promote beneficial changes in the gut microbiota, supporting the body’s own endogenous processes to decrease inflammation and increase healing. This concept of microbial alteration falls under the definition of regenerative medicine and should be included accordingly. By implementing microbial alterations in regenerative medicine, this current study could lend increasing support to the current research on the associations of the gut microbiota, bone metabolism, and AI-related musculoskeletal pathology.

## 1. Introduction

AI diseases represent a family of more than eighty disease processes affecting tens of millions of Americans. Women are affected disproportionately to men, and the rates of incidence are rising most rapidly in developed nations [1]. These diseases are widely varied in their pathologic characteristics, yet they all arise from a common self-reactive, adaptive immune response against the body’s tissues. Additionally, treatment of rheumatic disorders is influenced by each human’s microbiome or microbiota, the unique microbial environment seen in each individual person, but this interplay can be challenging due to variable and unpredictable responses to therapies [2]. In addition, the AI disease presentation can be vague and insidious, often irreversibly progressive by the time a clinical diagnosis can be assigned [1]. Because of this, the development of more effective early intervention strategies is imperative in improving disease management. Furthermore, many AI diseases are associated with chronic inflammatory states, which may lead to bone loss resulting from dysregulated bone remodeling pathways [3]. Considering that human gastrointestinal microbiota plays an integral role in regulating bone homeostasis [3], a connection could be made to assess the viability of modulating and restoring the gut microbiome as a form of regenerative medicine (RM) and a potential adjunct solution in the treatment of the autoimmune associated bone and joint pathologies. We use the terms microbiome and microbiota interchangeably throughout this manuscript, and we address the specific AI pathology of psoriatic arthritis, rheumatoid arthritis, inflammatory bowel disease, and systemic lupus erythematosus. 

## 2. Concept of Regenerative Medicine

Regenerative medicine (RM) is a rapidly developing interdisciplinary field that approaches treating multifactorial disease processes with convergent technological methods that simulate intrinsic biological repair mechanisms [4,5,6]. RM as a practice (a) emphasizes interventions that support the body’s own endogenous repair mechanisms and (b) attempts to better traditional replacement therapies by accounting for compounding physiologic and environmental contributions to disease processes [4,5,6]. The role of RM is to restore and rejuvenate impaired physiologic function by working with the body’s systems instead of pharmacologically silencing the body’s natural healing systems as has been traditionally done. RM is not a classical surgical intervention, a prosthetic replacement, or a pharmaceutical therapy (Figure 1). Still, the definition should not be so specific as to limit its concepts to stem cells, genetic engineering, autologous or allogenic biomolecules, tissue transplantation, or tissue engineering [4,6]. RM is an inclusive approach to using emerging technologies in an appropriate context to restore human form and function [6].

As recently as 2022, Preethya et al. published works addressing the effect of the gut microbiome in cell-based therapies and discussed its role as a tool in RM [7]. By all established parameters, microbiome alteration in the form of dietary and lifestyle changes, pre-and probiotic supplementation, and fecal microbiota transplantation is well suited to the definition of RM [7,8,9,10]. There is no clear separation of the gut microbiome and the endogenous human physiology, as their structure and function are intimately related. These commensal organisms produce short chain fatty acids that regulate intestinal cell growth and differentiation, inducing the secretion of mucin and other antimicrobial peptides [11]. The gut microbiome performs essential metabolic function via the synthesis of vitamins, amino acids and cholesterol, fermentation of non-digestible substrates, and the transformation of bile [11]. These microflorae also provide an essential protective role as they synthesize antimicrobial compounds, outcompete pathogenic organisms for nutrients and attachment sites; in addition, direct signaling interactions between the gut microbiome and the adaptive immune systems exist [11]. If RM emphasizes interventions which support the body’s own physiologic repair mechanisms and attempts to better account for the multifactorial causes of disease, then we would advocate that clinical alteration of the gut microbiome has both primary and adjunct therapy promise for RM. Additionally, these methods of therapeutic intervention may prove more accessible than other forms of RM, such as stem cell therapy, genetic engineering, or tissue engineering [8,9] (Figure 1).

## 3. Disease Characteristics

### 3.1. Psoriatic Arthritis

Psoriatic arthritis (PsA) is an inflammatory, immune-mediated arthropathy affecting joints and entheses. Despite this clear definition, the condition is diverse in its clinical presentation, including features such as nail disease, dactylitis, uveitis, and osteitis. Furthermore, recent epidemiologic studies have supported the idea that PsA is associated with various comorbidities, including cardiovascular disease, diabetes, hypertension, dyslipidemia, and cerebrovascular accidents [12]. The variety of comorbidities reflects that PsA is a multifactorial disorder that remains poorly understood; however, the current thought is that the pathogenesis of PsA is comparable to that of plaque psoriasis in that certain stressors may activate a systemic inflammatory response in susceptible individuals. Neutrophils, macrophages, mast cells, T cells, and dendritic cells are found to infiltrate the points of enthesis (sites of tendon insertion into bone and/or ligaments) and within the joints, creating a pannus of inflammatory mediators [12]. Additionally, many proinflammatory cytokines, including tumor necrosis factor (TNF)-a and interleukins (IL)-1, IL-9, IL-17, IL-18, IL-22, and IL-23, were found to be upregulated in the skin, synovium, and synovial membranes of PsA patients [12] (Figure 2). High-resolution imaging has also provided insight into specific bone formation phenotypes seen in PsA, suggesting the IL-23/IL-17 pathway is imperative to psoriatic bone remodeling [13]. In addition, the upregulation of these inflammatory cytokines has been linked to alterations in PsA patients’ microbiome.

Early identification of the disease and accurate diagnosis of specific patterns and types of PsA is crucial to preventing chronic structural damage that leads to patient disability and associated consequences. While the plethora of molecular and immune changes would facilitate diagnoses of PsA, current diagnosis is based on history, physical examination, the absence of rheumatoid factor, and specific radiographic features [14]. The use of screening tools, meticulous history intake, and a thorough physical exam can provide clear support for a PsA diagnosis and determine a specific presentation pattern. The goals of treatment for PsA are to achieve clinical remission, improve patients’ quality of life, and limit the extent of the disease. Achieving these treatment goals may require consideration of patient and disease-specific (i.e., severity) factors, especially when including pharmacological options such as disease-modifying antirheumatic drugs (DMARDs) [15]. There are current pharmacologic therapies developed to target specific cytokines and signaling pathways seen in PsA; however, at least 40% of patients only have a partial response or fail to respond to such treatments [16]. For this subpopulation of individuals, focusing on modifications to the microbiome may also be a potential treatment target. 

### 3.2. Rheumatoid Arthritis

Rheumatoid arthritis (RA) is a symmetric, inflammatory, peripheral polyarthritis of unknown etiology. It typically leads to deformity through the stretching of tendons and ligaments and the destruction of joints through cartilage and bone erosion. If left untreated or unresponsive to therapy, joint inflammation and destruction can lead to loss of physical function and inability to carry out activities of daily living. Uncontrolled inflammation may have other health risks, including higher rates of cardiovascular disease and osteoporosis [17]. As part of the pathogenesis of RA, anti-citrullinated peptide antibodies and the rheumatoid factor are thought to play a significant role, and so is the T- and B-cell activation. There also appears to be a decrease in the number of regulatory T (Treg)-cells, which may also play an essential role in the pathophysiology of the disease [18]. Additionally, cytokines have a crucial role in the pathophysiology of RA as pro-inflammatory cytokines, such as TNFα, IL-1, and IL-6, stimulating inflammation and degradation of bone and cartilage (Figure 2). Multisystem immune complications result from an imbalance between pro- and anti-inflammatory cytokines.

Alteration of T-cell differentiation by the gut microbiota and its metabolic products is one of the most abundant environmental factors connected to AI [19]. Rapid advances in the past decade have shown that dysbiosis of the gut microbiome is a key hallmark of rheumatoid arthritis (RA) [20]. The gut microbiota has been proposed as an indispensable environmental factor in the progression of RA [19]. Decreased gut microbial diversity of RA patients is associated with disease duration, and the expansion of rare microbial lineages characterizes the RA-associated gut microbiota [20]. The progressive understanding of the dynamic interaction between gut microbiota and their host may help in establishing the diagnosis and personalized management for each RA patient, thereby achieving better efficacy in clinical practice [21].

The diagnosis of RA can be established in a patient with inflammatory arthritis involving three or more joints, positive rheumatoid factor (RF) and/or anti-citrullinated peptide/protein antibody (ACPA), disease duration of more than six weeks, and elevated C-reactive protein (CRP) or erythrocyte sedimentation rate (ESR), but with no evidence of alternative diagnoses [22]. The diagnosis of RA can also be established in patients who do not meet all these criteria (e.g., patients negative for RF and ACPA). RA may be diagnosed without positive serologic results when there is a large number of swellings in a symmetrical pattern of small joints, or when there are erosions or rheumatoid nodules present [17]. Patients with these findings/clinical features are generally consistent with those described as meeting the American College of Rheumatology (ACR)/European Alliance of Associations for Rheumatology (EULAR) classification criteria for RA [22].

The current recommendations address treatment with conventional synthetic disease-modifying antirheumatic drugs (csDMARDs), biologic DMARDs (bDMARDs), targeted synthetic DMARDs (tsDMARDs), and glucocorticoids [23]. Selection of pharmacologic treatment is dependent upon comorbidities, patient preferences, regulatory guidance, and disease severity. Adding nonsteroidal anti-inflammatory drugs (NSAIDs) and/or glucocorticoids (systemic and/or intraarticular) for initial symptomatic control of inflammation while awaiting the response to DMARD therapy may also be helpful [23]. Alternatively, non-pharmacological therapy and atherosclerotic cardiovascular disease (ASCVD) prevention measures are associated with improved outcomes in patients with RA [24]. There is evidence that potentially anti-inflammatory diets (Mediterranean, vegetarian, vegan, ketogenic) result in significantly lower pain levels than ordinary diets [24].

### 3.3. Inflammatory Bowel Disease

Inflammatory bowel disease (IBD) is an autoimmune gastrointestinal disorder divided into Crohn’s disease (CD) and ulcerative colitis (UC). Defining features include diarrhea, abdominal pain, rectal bleeding, and weight loss. However, ulcerative colitis is more likely to present with diarrhea and bleeding, while Crohn’s disease is more likely to present with watery diarrhea and vague symptoms [25], such as bloating, fatigue, etc. Extraintestinal effects of IBD most commonly present as musculoskeletal manifestations and arise in about 40% of patients [26]. 

A common consensus on the pathogenesis of IBD is that three major host compartments that function together as an integrated “supraorganism” (microbiota, IECs, and immune cells) are affected by specific environmental (e.g., smoking, antibiotics, enteropathogens) and genetic factors that, in a susceptible host, cumulatively and interactively disrupt homeostasis during one’s life and, in so doing, culminate in a chronic state of dysregulated inflammation, i.e., IBD [27]. Based on genome-wide association studies, IBD is likely a polygenic process, and T-cell-mediated responses are amplified in UC and CD. The pathogenesis of IBD is also associated with microbial dysbiosis [27]. In fact, medications that have been found to alter the gut microbiome, such as antibiotics, NSAIDs, contraceptives, and statins, have all been shown to increase the risk of developing IBD [25]. 

The primary diagnostic modality for IBD is endoscopy, particularly colonoscopy, which may include multiple biopsies. Endoscopic findings may show loss of vascularity, erythema, erosions, ulcers, and pseudopolyps in the mucosa, with most findings in UC being in the colon and most findings in CD being in the colon and ileum with skip lesions throughout [25]. Biopsy results differ in UC vs CD. Crypt abscesses, mucin depletion, and thickened muscularis are found in UC biopsy specimens, while granulomas, fibrosis, and normal mucin can be found in CD biopsy specimens [26]. 

While the diagnosis of IBD may be relatively straightforward, its treatment remains challenging. The treatment of IBD can be divided into medical and surgical management. Medical treatment goals are early induction of remission and maintenance, since curative medical therapy has not been established [27]. UC treatment to achieve remission can vary based on the severity of symptoms but includes 5-aminosalicylic acid (5-ASA) as the first-line treatment, with additions of steroid enemas and oral prednisolone if needed (in that order). If a patient is steroid-dependent to maintain remission, it is recommended that immunomodulators, such as azathioprine and 6-mercaptopurine (AZA-6MP), be used. Crohn’s disease treatment can include an oral steroid and 5-ASA with enteral nutrition. Once the patient is in remission, AZA-6MP can be combined with enteral nutrition. Intravenous steroids and nutrition are highly recommended in severe or fulminant Crohn’s disease. With the challenge of treating IBD and the building evidence of the microbiome influencing IBD severity and extraintestinal manifestations of the disease, the microbiome could be a promising target for new therapies.

### 3.4. Systemic Lupus Erythematosus

Systemic Lupus Erythematosus (SLE) is an autoimmune disorder that affects various organ systems. The etiology of SLE is multifactorial and unknown; however, multiple factors impact the morbidity and mortality of the disease, such as genetic, immunological, endocrine, and environmental factors [28]. These factors result in a dysregulation of the immune system leading to pathogenic antibodies that target the host tissue, resulting in tissue damage. SLE is thought to occur due to a dysregulated adaptive immune response, where T and B cells are activated by host antigens and cause defective apoptotic pathway activation, resulting in increased and inappropriate antibody production [29]. SLE presents numerous detectable antibodies such as anti-nuclear antibodies (ANAs), anti-dsDNA, anti-Smith, anti-La, and anti-Ro [30]. SLE also has many dysbiosis similarities compared to IBD, diabetes mellitus, multiple sclerosis, and RA [29]. Unfortunately, there have not been extensive studies comparing AI and the causes of their shared dysbiosis characteristics.

SLE is diagnosed based on a combination of clinical manifestations, histology of affected organs, and blood tests. In 85% of the patients, cutaneous lesions are the first symptoms. However, subtype categories of SLE do exist. In 90% of the SLE patients, constitutional symptoms are usually the initial presenting feature. Fatigue, malaise, fever, anorexia, and weight loss are common. Due to the immunocompromised state of the SLE patients, infectious causes must always be ruled out during flare-ups in these patients while working up the fever. Furthermore, SLE is a very rare cause of fever of unknown origin [28]. Additionally, anemia, thrombocytopenia, and leukopenia are common findings in SLE patients’ blood tests [30]. 

Moreover, there are musculoskeletal factors affected by SLE. Approximately 80% to 90% of patients with SLE suffer from musculoskeletal involvement at some point during their disease course, and symptoms may range from mild arthralgias to deforming arthritis [29]. Lupus arthritis is an inflammatory polyarthritis that can involve any joint but mainly targets the small joints of the hands, knees, and wrists. More specifically, Jaccoud arthropathy is a condition associated with SLE that can mimic RA, presenting with metacarpophalangeal joint subluxation and ulnar deviation due to joint capsule and ligament laxity [25]. Another condition that can present with SLE is avascular necrosis, usually involving the hip joint, potentially bilaterally. Additionally, SLE often manifests with inflammatory myopathy. Lastly, SLE patients have a higher risk of developing fibromyalgia and RA nodules [29]. 

There are several treatment modalities for SLE. Pharmaceutical treatments include antimalarial drugs (chloroquine (HCQ)), corticosteroids (methylprednisolone, prednisone), immunosuppressants (AZA, mycophenolate mofetil (MMF), and cyclophosphamide (CYC)), as well as biologicals (belimumab, rituximab) [30]. When patients are using any of these pharmacological treatments, they need to be tested for co-morbidities. Other pharmaceutical treatments include DMARDs such as MTX, leflunomide, and calcineurin inhibitors, or biologics such as TNF inhibitors, abatacept, and tocilizumab [29]. Finally, for non-pharmacological treatment, changing lifestyle factors, such as increasing physiotherapy or exercise, can effectively regulate inflammatory symptoms. Additionally, increasing probiotic use has been shown to reduce autoantibodies and decrease lupus severity [30]. 

## 4. Disease Summary

Psoriatic arthritis, Rheumatoid arthritis, Inflammatory Bowel Disease, and Systemic Lupus Erythematosus have all been shown to affect joint health. While many medications can be used to alleviate symptoms of these autoimmune conditions, they have various adverse effects, including gut microbial dysbiosis. Additionally, the nature of these diseases results in generalized symptoms such as chronic pain, fatigue, headache, etc. This often makes current treatment options limited as they tend to focus on a specific factor of disease, such as a certain inflammatory marker or a particular symptom. On the other hand, patients can be put on many medications to treat all their symptoms, potentially causing adverse effects. A consequence of our current treatment modalities can lead to patients on multiple medications or those with subtherapeutic results. AI has continuously shown its need for individualized treatments tailored to a patient’s specific physiologic needs. Regenerative medicine is a promising treatment option because it encourages the body to create and enhance the specific mechanisms it needs to heal. RM may also benefit various joint concerns caused by these diseases with fewer side effects than current medications being used. 

## 5. Microbiome Alterations Seen with Autoimmune Pathology

### 5.1. Psoriatic Arthritis

In addition to pharmacological and lifestyle considerations, there has also been an increased interest in the association between PsA and the gut microbiome, especially in the context of potential dysbiosis-focused therapies for PsA. The association between dysbiosis and psoriatic states is only just emerging and merits further research. However, it is thought that decreased beneficial taxa and overall microbial diversity impair the intestinal barrier, allowing bacteria to migrate through the intestinal wall and spread to other body systems. This has been shown to increase various immunological markers, such as IL-23, IL-17, IL-22, and TNF-a, leading to a general inflammatory state and encouraging the development of psoriatic lesions, PsA, and several other pathologies (Figure 2). In a 2015 study published by the American College of Rheumatology, Scher et al. determined that PsA patients had a lower relative abundance of multiple intestinal bacteria compared to healthy controls and a lower abundance of reportedly beneficial taxa, specifically *Akkermansia, Ruminococcus,* and *Pseudobutyrivibrio* [31]. Interestingly, the local gut immune response in PsA is characterized by a decrease in RANKL, and increased fecal sIgA levels relative to healthy control; however, it is still unclear if dysbiosis precedes or follows the onset of arthritis and joint damage [32].

Additionally, PsA samples showed similar microbial patterns as those seen in IBD samples with changes in specific inflammatory proteins that were distinct from patients with skin psoriasis and healthy controls. In both disease states, *F. prausnitzii*, *Bifidobacterium* spp., *Lactobacillus* spp., *Parabacteriodes*, and *Coprobacillus* were underrepresented. At the same time, the presence of *Salmonella* sp., *Campylobacter* sp., *Helicobacter* sp., *Escherichia coli, Alcaligens* sp., and *Mycobacterium* sp. increased [33]. An additional study published in the *International Journal of Molecular Science* in 2021 found that PsA patients have a decreased overall diversity in microbiota. PsA patients had reduced representation of what is considered beneficial bacteria, such as *Cutibacterium, Burkholderia* spp. and *Lactobacilli*, and increased representation of *Corynebacterium kroppenstedii, Coyrynebacterium simulans, Neisseria* spp., and *Finegoldia* spp. associated with psoriatic skin compared to healthy controls [34]. Lastly, Ciccia et al. found that IL-9, a known survival factor for pathologic T cells located in the synovium of inflammatory arthritis, was specifically overexpressed in PsA patient’s gut microbiome and their synovial tissues [35]. Isolated epithelial cells from PsA gut tissues also showed increased IL-23p19 mRNA levels when stimulated with IL-9 in vitro, indicating a potential relationship between microbial dysbiosis and altered production of IL-9, IL-23, and IL-17 with potential to circulate joint tissues and promote arthritis [32].

While some studies have focused on specific microbiota and tried to define the ratio of beneficial to harmful taxa, other studies have focused on the diet itself being a possible contributing factor to the pathogenesis of PsA. In January 2021, Shi et al. found that a high-sugar and moderate-fat diet (i.e Western diet) promoted IL-23-mediated skin inflammation and infiltration of γδ T cells as well as enhanced expression of Th17 cytokines and susceptibility to IL-23 mediated joint inflammation [36]. To further establish the relationship between the Western diet and its pro-inflammatory effect, the study also showed that when fat and sugar intake was reduced, it decreased IL-23-mediated inflammation and proliferated microbial diversity, including increased *Bacteroidetes* and decreased *Proteobacteria* and a subset of *Firmicutes* [36]. This suggests that the Western diet could be an additional factor causing the dysbiosis in PsA patients, enhancing the pro-inflammatory state throughout the body.

### 5.2. Inflammatory Bowel Disease

Lifestyle changes likely play a significant role in treating IBD, with emerging new research regarding nutrition and dysbiosis and their role in IBD [15]. Currently, the pathogenesis of IBD is thought to include susceptible individuals that are exposed to gut antigens, such as microbiome changes which result in inappropriate activation of the immune system [37]. When experimental murine models of IBD are examined, under germ-free conditions, such as in the Britton et al. experiment, IBD is either not developed at all or significantly improved based on the expectation, which is highly suggestive of the role microbes play in IBD intestinal inflammation [38]. Further, alterations in microbiome components can be involved in different IBD subtypes. In a large study using 1815 stool samples from 303 patients with CD, 228 patients with UC, and 161 healthy controls, alpha diversity, meaning the observed taxa or evenness of a sample, was significantly lower in patients with both CD and UC compared with active controls, and active UC showed significantly lower alpha diversity than when stool was taken from the same subject during a time of inactive UC [39]. In other studies with IBD patients, there were notable decreases in the relative abundances of *Firmicutes* and an increase in *Proteobacteria* [40]. The *Enterobacteriaceae* family is increased in the gut of IBD patients, a trend that is even higher in IBD patients with arthropathy [40]. According to Longman et al., it was found that IgA-coated *Escherichia coli*, part of the *Enterobacteriaceae* family, are enriched in patients with Crohn’s disease (CD)-associated spondylarthritis compared to patients with CD only [40]. Hence, extraintestinal manifestations of IBD may arise from the cross-reactivity of antigen-specific immune responses against intestinal antigens happening at non-intestinal sites. If the microbiome changes antigen-specific immune responses at joints, targeting the microbiome could be a promising new target for treating arthritis and other joint issues associated with IBD.

### 5.3. Rheumatoid Arthritis

RA patients have been shown to have alterations in the composition of the gut microbiota. Alterations of the gut microbiota that are associated with rheumatoid arthritis are likely multifactorial. Some proposed mechanisms include activation of antigen-presenting cells through an effect on toll-like receptors (TLRs) or nucleotide-binding oligomerization domain-like receptors (NLRs), ability to produce citrullination of peptides by enzymatic action, antigenic mimicry, alterations in permeability of intestinal mucosal, control of host immune system (triggering T-cell differentiation) and increase of T helper type 17-mediated mucosal inflammation [41].

A study by Bennike et al. [42] showed 21 citrullinated peptides in colonic tissue from RA patients as well as controls; they have previously been found in lung tissue and synovial fluid from RA patients. Three citrullinated proteins (citrullinated vimentin, fibrinogen-alpha, and actin) are known targets for anticitrullinated protein antibodies (ACPAs), supporting the assertion that colon mucosa could be a potential break site for immune tolerance towards citrullinated epitopes. Compared to the controls, citrullinated vimentin was found in greater abundance in the colonic tissues of these patients, suggesting the initial trigger of rheumatoid arthritis does not reside in just one specific area of the body, but can also take place in many other locations [42]. The results of this study support the hypothesis that the mucosa of the colon may act as a break site for immunity to citrullinated proteins, triggering ACPA production in patients with impaired immunity [41].

A recent study collected fecal samples from 205 RA patients and 199 healthy controls (HCs) for bacterial DNA extraction and 16S ribosomal RNA (rRNA) gene sequencing [43]. The gut microbiome results from RA patients were significantly different from those of HCs. Abnormal bacterial communities are associated with altered levels of lymphocyte subpopulation and cytokines, a potential pathogenesis of RA [43]. Furthermore, variations in the microbiota constituent populations and their frequency (i.e., dysbiosis) can stimulate several autoimmune and inflammatory pathologies through a loss of equilibrium in the T cell subpopulation (e.g., Th1, Th2, Th17, and Treg cells) [44]. Among several inflammatory cells, Th17 cells producing IL-17, IL-21, and IL-22 have been identified as the major pathological factor in the exacerbation of RA [35] (Figure 2). Some of these microbial variations include a lower abundance of *Bifidobacterium* and *Bacteroides* in patients with RA. At the same time, the populations of *Lactobacillus salivarius*, *Lactobacillus iners,* and *Lactobacillus ruminis* are higher in patients with early RA [19]. *Prevotella copri* is also positively correlated with new-onset, untreated RA patients [19]. A study recognized that *P. copri* had a notable ability to stimulate the production of Th17 cell-related cytokines, specifically IL-6 and IL-23, which increase mucosal inflammation [45]. A relative increase in *Collinsella* was also identified in patients with RA. *Collinsella* causes disease by enhancing gut permeability, as seen by decreased tight junction protein expression, and affects the epithelial release of IL-17A [45]. 

Despite clear evidence of alterations in the gut microbiota in the context of RA, the therapeutic effects of probiotic bacteria in rheumatoid arthritis (RA) remain controversial. A recent study indicates there may be some evidence for therapeutic effects in RA patients [46]. Significant differences in gut microbiota composition were observed in patients with different RF levels. The relative abundance of *Bifidobacterium* and *Collinsella* was lower in RF-high when compared to RF-low and RF-negative RA patients. In comparison, the relative abundance of *Clostridium* and *Ruminococcaceae* was higher in RF-high patients when compared to RF-low and RF-negative patients. Among 10 differentially abundant *Bifidobacterium*, the *B. longum* RAPO strain exhibited the most substantial ability to inhibit IL-17 secretion. Oral *B. longum* RAPO was provided in a murine model, with findings suggesting that *B. longum* RAPO may alleviate RA by inhibiting the production of IL-17 and other proinflammatory mediators. The most important limitation of this study is the lack of evidence on humans. Although *B. longum* RAPO exerted strong anti-RA effects in mice, these effects should be confirmed in human studies [46]. 

### 5.4. Systemic Lupus Erythematosus

In SLE, the gastrointestinal system influences and is influenced by the pathophysiology of the disease. The presentations can appear anywhere on the GI tract and include esophageal dysmotility (especially the upper one-third part of the esophagus), mesenteric vasculitis, lupus enteritis, peritonitis and ascites, protein-losing enteropathy, pancreatitis, and lupoid hepatitis. In a study conducted by Zhang et al., the authors found an overrepresentation of *Lachnospiraceae* in the gut biome of female mice that correlated with a more severe set of lupus symptoms [28]. Moreover, high levels of Clostridiaceae and Lachnospiraceae have been shown to reduce the time progression of the disease [47]. Additionally, *Lactobacillus rhamnosus* and *delbrueckii* have been studied for probiotic effects due to their ability to obtain nutrients and act as anti-inflammatory agents. Hence, increasing *Lactobacilli* content can correct leaky gut, promote IL-10, improve renal function, and prolong mouse survival [28]. Furthermore, in a study by Hevia, human stool samples were analyzed from 40 patients, 20 patients with SLE and 20 patients as control group. The authors found that SLE patients’ stools showed decreased Firmicutes to Bacteroidetes ratios. They also found that the reduced ratio is associated with increased oxidative phosphorylation in the intestinal microbiota [48]. Furthermore, according to Zang et al., caloric restriction has been shown to promote intestinal gut microbiota change and reduce SLE disease progression [47].

## 6. Discussion: Using the Microbiome as a Treatment Target

As evidenced through the examination of each AI, the microbiome plays a vital role in the pathogenesis and course of several AI disease types. PsA and IBD share many of the same shifts in the microbiota, such as higher levels of *Salmonella, Campylobacter, Helicobacter, Escherichia coli, Alcaligens*, and Mycobacteria (Figure 3). Additionally, both PsA and IBD demonstrated multiple suppressed species, such as *Prausnitzii*, *Bifidobacterium*, *Lactobacillus*, *Parabacteriodes*, and *Coprobacillus* when compared to healthy individuals (Figure 3). In RA patients, gut dysbiosis stimulates the cardinal inflammatory pathways caused by T-cell destabilization. In SLE, an increase in populations of *Lachnospiraceae* was correlated with more severe symptoms (Figure 3), while increased *Lactobacilli* species was shown to decrease gastrointestinal symptoms associated with SLE and promote the anti-inflammatory effects of IL-10. Fecal microbiota transplantation or inoculation with specific microbes in animal models of AI support the hypothesis that alterations of gut microbiota influence autoimmune responses and disease outcome [49]. Even in AI without closely shared microbiome characteristics, there is a common thread of pathogenic bacterium being able to induce inflammatory pathways while suppressing the anti-inflammatory functions of symbiotic bacterial groups.

As understanding of the gut microbiome and its role in human health continues to develop, it should be considered as part of the valuable practice of regenerative medicine. RM is an attempt to emphasize medical interventions which support the body’s repair mechanisms. The aim is to improve traditional therapies by better accounting for the compounding factors and complexities of human disease processes. Modifying and encouraging a healthy and diverse gut microbiota includes a spectrum of practices, from targeting a specific bacterial group to produce a desired outcome to effecting beneficial changes to one’s diet and lifestyle.

When analyzing specific mechanisms to alter the microbiome in the existing efforts, a potential focus of treatment target could be altering the microbiome via macronutrient intake to decrease gut permeability associated with AIs. The monolayer of intestinal epithelial cells forms tight junctions (TJs), whereas the mucosal barrier contains mucin, antimicrobial peptides, and IgA secreted by goblet cells, Paneth cells, and plasma cells, respectively [49]. The intestinal mucosa typically serves as a multilayered barrier, restricting the permeation of external antigens and the leakage of endogenous contents. However, the Western diet reduces the abundance of *Bacteroidetes* and reciprocally enhances the abundances of *Firmicutes* and *Proteobacteria*, increasing the ratio of mucin-degrading bacteria [49]. After reducing these specific bacteria, the mucin layer is decreased, allowing pathogenic bacteria to translocate through the intestinal mucosa. Rebalancing this ratio could potentially reduce gut permeability, decreasing bacterial translocation.

In 2021, Matei et al. assessed K/BxN mice with spontaneous arthritis, wild-type, genetically modified (IL)-10R^−/−^, and claudin-8^−/−^ mice with induced arthritis, finding that arthritic mice displayed increased gut permeability, bacterial translocation, inflammatory gut damage, and increased interferon γ, with decreases in IL-10^+^ intestinal-infiltrating leukocyte frequency, and reduced intestinal epithelial IL-10R expression [50]. This suggests that increased intestinal permeability contributes to arthritis development. Targeting bacteria that are intimately affiliated with gut permeability could be a potentially alternative pathway in microbial alteration leading to decreased gut permeability.

Additionally, targeting a pathogen’s unique enzymes could potentially be a pathway to creating new treatment options for an AI disease. Recently, Mills et al. analyzed six fecal- or serum-based omic datasets from a cohort of 40 UC patients as well as a validation cohort of 210 samples (73 UC, 117 Crohn’s disease, and 20 healthy controls), and determined that both cohorts showed a subset of clinically active UC patients had an overabundance of proteases originating from *Bacteriodes vulgatus* [51]. Furthermore, the study found that a broad-spectrum protease inhibitor improved *B. vulgatus*-induced barrier dysfunction in vitro, and prevented colitis in *B. vulgatus* monocolinized, IL-10 deficient mice [51]. This study shows a promising link between targeting specific pathogens linked with certain AI disease; however, the current literature discussing these connections is limited and warrants further investigation.

As the beneficial effects of microbiome alteration in clinical settings emerge from research studies and clinical trials, so does the spectrum of possible deleterious implications in microbiome intervention. Though microbiome alteration is challenging to isolate as a specific causative mechanism for adverse events, it is imperative to understand the fundamental risk to the therapy period. The following is a discussion of emerging efforts to understand complications of microbiome alteration, which can be particularly pertinent for chronically ill and immunocompromised patients.

While probiotic supplementation is extremely well tolerated by the general population, there are numerous case reports of systemic infections, such as bacteremia and endocarditis, which were suggested to have arisen secondary to probiotic therapy and persons with severe underlying disease [52,53,54]. In several instances, patients suffering from severe pancreatitis that were clinically supplemented with probiotics, specifically *Lacticaseibacillus* spp, showed a higher mortality rate due to small bowel ischemia versus patients that were not treated with probiotics [53,55]. There exists a theory that probiotics can induce innate and adaptive immune reactions as we introduce non-self-organisms into the body; however, no direct connections of probiotic-induced autoimmunity reactions have been established [52,54].

Another factor to consider when discussing microbiome alteration is how these changes can affect pharmacologic treatment options. Interactions between the gut microbiome and pharmaceutical drugs are known as pharmacobiomics [56]. In pharmacobiomic reactions, gut microbiota can directly influence an individual’s response to a specific drug by enzymatically transforming the drug structure and altering its bioavailability, bioactivity, or toxicity. Clearly stating, altering the microbiome could lead to potentially beneficial or harmful effects depending on the pharmacobiomics. For example, in multiple studies, it was shown that Lactobacilli strains contain genes which demonstrate potential resistance to common antibiotics and some lateral genetic transfer, which has been induced in vitro; however, no clinical evidence of this genetic transfer has been shown in humans [52,54]. On the contrary, a recent study found that Metformin, a popular drug used to treat type 2 diabetes mellitus, induces gut microbiome producers of short chain fatty acids, which positively affects the drug performance; however, Brivudine, an antiviral drug commonly used to treat herpes zoster, is converted to bromovinyluracil, a potentially hepatotoxic compound by the gut microbiome [56].

Additionally, it is worth discussing how current AI pharmacologic therapies could potentially affect the microbiome itself in an AI disease. In 2021, Nayak et al. found that RA patient microbiotas had distinct changes in gut bacterial taxa between MTX responders and non-responders, and transplantation of post-treatment samples into germ-free mice given an inflammatory trigger led to reduced immune activation relative to pre-treatment controls, enabling identification of MTX-modulated bacterial taxa associated with intestinal and splenic immune cells [57]. This suggests that MTX largely alters the microbiome, indicating that there is potential for other pharmacological treatments to do the same. Further research is needed to show the potential consequences of these effects, as well as the way other pharmacological options affect the gut microbiota.

When assessing whether microbiome alteration is a worthy alternative to the existing drugs or a potential origin for new drugs, current studies show that there is a potential for developing new treatments from the taxa themselves. Manipulation of the intestinal microbiome has great potential to reduce the incidence and/or severity of a wide range of human conditions and diseases, and the biomedical research community now faces the challenge of translating our understanding of symbiotic microbial species into beneficial medical therapies [58]. While there is still much to uncover in the literature regarding the biomedical application of the microbiome, current therapies implementing this have shown to represent true promise. For example, fecal microbiota transplantation (FMT) has proven to be an effective treatment of gastrointestinal disorders caused by microbial disbalance and is now being used to treat extra-gastrointestinal conditions such as metabolic and neurological disorders, which are considered to have their provenance in microbial dysbiosis in the intestine [59]. While studies and current treatment modalities implementing microbiome alteration as therapy have been increasing, modifying gut microbiota as a form of medication is a potential that has not been earnestly appreciated. This justifies the need for continual research in the field.

The evidence discussed in this review suggests a strong connection between gut dysbiosis in autoimmune disorders and their musculoskeletal symptoms of joint degeneration. We propose the value of including gut microbiome as a form of regenerative medicine is worthy of continued investigation to alleviate these symptoms of AI diseases. Additionally, further studies are needed to delve into the true intricacies that microbiome alteration demands, as well as whether the use of microbes represents a substantial alternative to the existing commercial drugs or as a potential new source for pharmacotherapy. We appreciate the briefness of this discussion; however, this reflects the fact that microbiome alteration as a potential treatment strategy for AI diseases is not heavily relied upon. Based on the analysis of the current literature and content regarding AI diseases, this approach should be considered in future treatment.

## Figures and Tables

**Figure 1 biomedicines-11-01582-f001:**
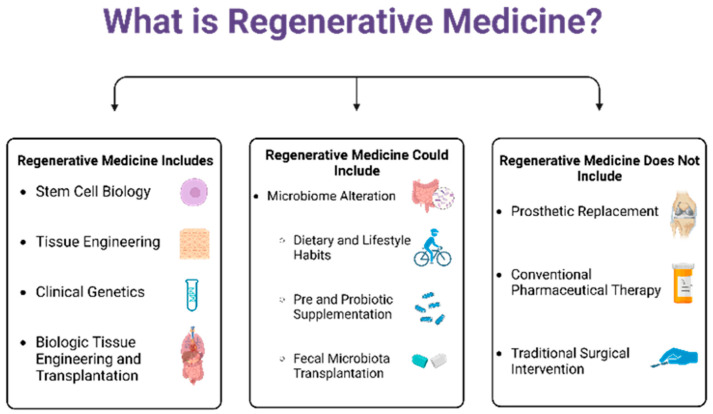
Current Therapies Included in and Excluded from the Definition of Regenerative Medicine and Possible Adjunct Therapies to Be Included.

**Figure 2 biomedicines-11-01582-f002:**
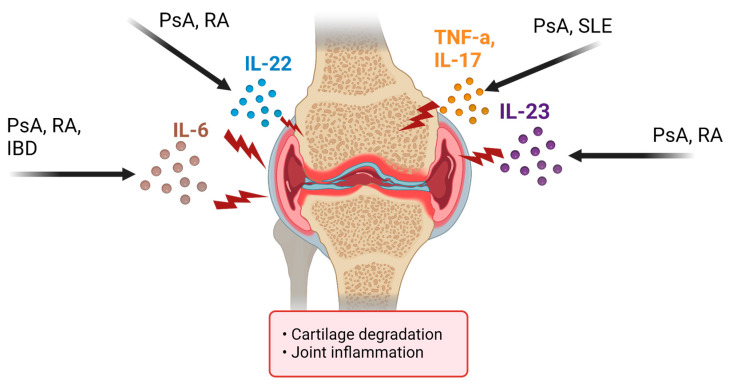
Common Interleukin Overlap Within AI Contributing to Joint Arthropathy.

**Figure 3 biomedicines-11-01582-f003:**
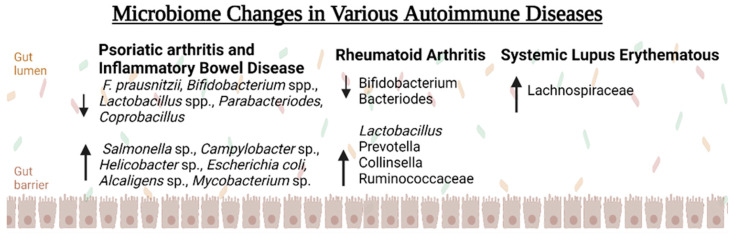
How the Gut Microbiome is a Mediator in Bone Metabolism.

## Data Availability

Data sharing is not applicable.

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
