# Peer review of "Using the Microbiome as a Regenerative Medicine Strategy for Autoimmune Diseases"

_biomedicines, 2023, doi:10.3390/biomedicines11061582_

Round 1

Reviewer 1 Report

This manuscript investigated the role of microbes in regenerative medicine. This article claims that using of gut microbiota could be a suitable  joint destruction in autoimmune disease Therefore, I suggest a minor correction and require a detailed clarification. Correction to be addressed by the authors as follows: The abstract is not well organized, where the sentences are incomplete and no continuity is there. It would be feasible, if include the significance of the current study in the abstract. A brief description of how the authors selected information from the literature in the databases, as well as doses.
Authors should justify and expand the information on the biomedical application of microbes , highlighting the main contribution in in vitro fields. Authors should specify the main experimental conditions used on the evidences from the literature. Where they briefly describe the most important data reported in the literature in a homogeneous manner and sequence reinforcing the relevance of microbiota  as medicinal alternative.
The most significant  mechanism of action of these microbes should be described and noticed more emphatically. Authors should discuss whether the use of microbes represents a solid alternative to existing commercial drugs or a source of new drugs.
Please add below studies to your manuscript in discussion section and also please discuss about possible side effects of microbes.
Conclusions should reaffirm the fundamental contribution of this paper.

Author Response

Response to Reviewer 1 Comments

This manuscript investigated the role of microbes in regenerative medicine. This article claims that using gut microbiota could be a suitable joint destruction in autoimmune disease. Therefore, I suggest a minor correction and require detailed clarification. Correction to be addressed by the authors as follows: The abstract is not well organized, where the sentences are incomplete and no continuity is there. It would be feasible, if include the significance of the current study in the abstract. A brief description of how the authors selected information from the literature in the databases, as well as doses.

Response 1: The abstract has been reconstructed with additional material included to enhance the cohesiveness of the subject discussed. Additionally, the significance of the current study has been included.

 “This review introduces the concept of altering an individual's specific gut microbiota as a form of regenerative medicine and potential tailored therapy for joint destruction seen in AI.  We hope to show multiple, specific ways to target the microbiome through diet changes, rebalancing microbial diversity or decreasing specific microbes associated with increased gut permeability, leading to reduced systemic inflammation contributing to joint pathology. Additionally, we plan to show how diet alterations can support beneficial changes in the gut microbiota, supporting the body’s own endogenous processes to decrease inflammation and increase healing. This concept of microbial alteration falls under the definition of regenerative medicine and should be included accordingly. By implementing microbial alterations in regenerative medicine, this current study could lend increasing support to the current research on the associations of the gut microbiota, bone metabolism, and AI-related musculoskeletal pathology.”

Authors should justify and expand the information on the biomedical application of microbes, highlighting the main contribution in in vitro fields. Authors should specify the main experimental conditions used on the evidence from the literature. Where they briefly describe the most important data reported in the literature in a homogeneous manner and sequence reinforcing the relevance of microbiota as medicinal alternative. The most significant mechanism of action of these microbes should be described and noticed more emphatically. Authors should discuss whether the use of microbes represents a solid alternative to existing commercial drugs or a source of new drugs.

Response 2: The information regarding the biomedical application of microbes as well as the use of microbes representing a solid alternative to existing commercial drugs or new source for potential drugs has been addressed.

“Additionally, it is worth discussing how current AI pharmacologic therapies could potentially affect the microbiome itself in AI disease. In 2021, Nayak et. al found how RA patient microbiotas had distinct changes in gut bacterial taxa between MTX responders and non-responders, and transplantation of post-treatment samples into germ-free mice given an inflammatory trigger led to reduced immune activation relative to pre-treatment controls, enabling identification of MTX-modulated bacterial taxa associated with intestinal and splenic immune cells [60]. This suggests that MTX largely alters the microbiome, indicating that there is potential for other pharmacological treatments to do the same. Further research is needed to show the potential consequences of these effects, as well as how other pharmacological options affect the gut microbiota as well.

When assessing if microbiome alteration is a worthy alternative to existing drugs or a potential origin for new drugs, current studies show there is a potential at developing new treatments from taxa itself. Manipulation of the intestinal microbiome has great potential to reduce the incidence and/or severity of a wide range of human conditions and diseases, and the biomedical research community now faces the challenge of translating our understanding of symbiotic microbial species into beneficial medical therapies [61]. While there is still much to uncover in the literature regarding the biomedical application of the microbiome, current therapies implementing this have shown to represent true promise. For example, fecal microbiota transplantation (FMT) has proven to be an effective treatment of gastrointestinal disorders caused by microbial disbalance and is now being used to treat extra-gastrointestinal conditions like metabolic and neurological disorders, which are considered to have their provenance in microbial dysbiosis in the intestine [62]. While studies and current treatment modalities implementing microbiome alteration as therapy have been increasing, modifying gut microbiota as a form of medication is a potential that has not been earnestly appreciated and justifies the need for continual research in the field.”

Please add below studies to your manuscript in discussion section and also please discuss about possible side effects of microbes.

Response 3: The possible side effects of microbes as well as additional content and references have been added to the discussion.

“As the beneficial effects of microbiome alteration in clinical settings emerge from research studies and clinical trials, so does the spectrum of possible deleterious implications in microbiome intervention. Though microbiome alteration is challenging to isolate as a specific causative mechanism for adverse events, it is imperative to understand the fundamental risk to the therapy period the following is a discussion of emerging efforts to understand complications of microbiome alteration, which can be particularly pertinent for chronically ill and immunocompromised patients.

Though probiotic supplementation is extremely well tolerated by the general population, there are numerous case reports of systemic infections, such as bacteremia and endocarditis, which were suggested to have arisen secondary to probiotic therapy and persons with severe underlying disease [47,48,49]. In several instances patients suffering from severe pancreatitis were clinically supplemented with probiotics, specifically Lacticaseibacillus spp, showed a higher mortality rate due to small bowel ischemia versus patients that were not treated with probiotics [48,50]. it has been theorized that probiotics can induce innate and adaptive immune reactions as we introduce non-self-organisms into the body, though no direct connections of probiotic induced autoimmunity reactions have been established [48,49]. Lactobacilli strains have genes, which demonstrate potential resistance to common antibiotics, and some lateral genetic transfer which has been induced in vitro, however no clinical evidence of this genetic transfer has been shown in humans [48,49].

An interaction between the gut microbiome and pharmaceutical drugs is known as pharmacobiomics [51]. In pharmacobiomic reactions, gut microbiota can directly influence an individual's response to a specific drug by enzymatically transforming the drug structure and altering its bioavailability, bioactivity, or toxicity. For example, metformin induces gut microbiome producers of short chain fatty acids which positively affects the drug performance however Brivudine is converted to bromovinyluracil by the gut microbiome which can induce hepatotoxicity [51].”

Conclusions should reaffirm the fundamental contribution of this paper.

Response 4: Conclusions in various sections of the paper have been augmented to reaffirm the contribution of this paper.

“This review aims to define mechanisms of microbiome crosstalk between the cells of bone and cartilage, as well as to investigate potential bidirectional connections between AI, bony and cartilaginous tissue, and the gut microbiome. By doing this, the review will also introduce the concept of altering an individual's specific gut microbiota as a form of regenerative medicine and potential tailored therapy for joint destruction seen in AI. We hope to show multiple, specific ways to target the microbiome through diet changes, rebalancing microbial diversity, or decreasing specific microbes associated with increased gut permeability, leading to reduced systemic inflammation contributing to joint pathology. Additionally, we plan to show how diet alterations can support beneficial changes in the gut microbiota, supporting the body’s own endogenous processes to decrease inflammation and increase healing. This concept of microbial alteration falls under the definition of regenerative medicine and should be included accordingly. By implementing microbial alterations in regenerative medicine, this current study could lend increasing support to the current research on the associations of the gut microbiota, bone metabolism, and AI-related musculoskeletal pathology.”

“If RM emphasizes interventions which support the body’s own physiologic repair mechanisms and attempts to better account for the multifactorial causes of disease, then we would advocate that clinical alteration of the gut microbiome has both primary and adjunct therapy promise for RM. Additionally, these methods of therapeutic intervention may prove more accessible than other forms of RM, such as stem cell therapy, genetic engineering, or tissue engineering [9,10].”

“The evidence discussed in this review suggests a strong connection between gut dysbiosis in autoimmune disorders and their musculoskeletal symptoms of joint degeneration. We propose the value of including gut microbiome as a form of regenerative medicine is worthy of continued investigation to alleviate these symptoms of AI disease. Additionally, further studies are needed to delve into the true intricacies that microbiome alteration demands, as well as if the use of microbes representing a substantial alternative to existing commercial drugs or as a potential new source for pharmacotherapy. We appreciate the briefness of this discussion; however, this reflects how microbiome alteration as a potential treatment strategy for AI disease is not heavily relied upon. Based on the analysis of current literature and content regarding AI disease, this approach should be considered in future treatment.”

Reviewer 2 Report

Abstract Line 17 and 18: The meaning of this statement that 'many autoimmune diseases share multiple inflammatory markers, there have been a significant number of disease-specific inflammatory markers than have shown to contribute to arthropathy that have been linked to certain levels and distributions of gut dysbiosis' Can the levels and distribution be specific?

The authors can consider to introduce the abbreviation of autoimmune diseases as AI in the first instance and use AI thereafter.

The author's need to introduce the significance or role of gut microbiota in a sentence and then connect it to gut dysbiosis and autoimmune diseases.

The concept of why gut microbiota is considered as regenerative medicine or an adjunct to it is not clear.

Author Response

Response to Reviewer 2 Comments

Abstract Line 17 and 18: The meaning of this statement that 'many autoimmune diseases share multiple inflammatory markers, there have been a significant number of disease-specific inflammatory markers than have shown to contribute to arthropathy that have been linked to certain levels and distributions of gut dysbiosis' Can the levels and distribution be specific?

Response 1: The statement has been reorganized and adjusted per the reviewer’s request. Additionally, the specific levels of inflammatory markers and how they affect gut dysbiosis is explained further in the disease-specific and disease summary paragraphs.

“Additionally, multiple AI’s have increased levels of certain inflammatory markers such as TNF-a, IL-6, and IL-17, that have been shown to contribute to arthropathy and also being linked to increased levels of gut dysbiosis.”

The authors can consider introducing the abbreviation of autoimmune diseases as AI in the first instance and use AI thereafter.

Response 2: The abbreviation of autoimmune diseases has been introduced as AI in the first instance and is used thereafter in the manuscript.

The author's need to introduce the significance or role of gut microbiota in a sentence and then connect it to gut dysbiosis and autoimmune diseases.

Response 3: The significance of the gut microbiota was introduced and the connection between gut dysbiosis and autoimmune diseases is discussed in lines 12-16.

“Gut microbiota is an integral aspect of human functioning, being intimately integrated into the metabolic, mechanical, immunological, and neurologic pathways of the body. The microbiota dynamically changes throughout our lifetimes and is individually unique. While the gut microbiome is ever-adaptive, gut dysbiosis can exert a significant influence on physical and mental health.”

The concept of why gut microbiota is considered as regenerative medicine or an adjunct to it is not clear.

Response 4: We appreciate the reviewer’s concern regarding why the gut microbiota being considered as regenerative medicine or as unclear. We have added clarification as to the concept of how gut microbiota can be considered a component of regenerative medicine and hope that we have created a more concise pathway between the ideas.

“There is no clear separation of the gut microbiome and the endogenous human physiology, as their structure and function are intimately related. These commensal organisms produce short chain fatty acids that regulate intestinal cell growth and differentiation, as well as inducing the secretion of mucin, and other antimicrobial peptides [46]. The gut microbiome performs essential metabolic function via the synthesis of vitamins amino acids and cholesterol, fermentation of non-digestible substrates, and the transformation of bile [46]. This microflora also provides an essential protective role as they synthesize antimicrobial compounds, outcompete pathogenic organisms for nutrients and attachment sites, and direct signaling interactions between the gut microbiome and the adaptive immune systems exist [46]. If RM emphasizes interventions which support the body’s own physiologic repair mechanisms and attempts to better account for the multifactorial causes of disease, then we would advocate that clinical alteration of the gut microbiome has both primary and adjunct therapy promise for RM.”

Reviewer 3 Report

First, the reviewer would like to appreciate the author's efforts to compile all this information to put forward the concept of altering the gut microbiome as a therapeutic strategy in autoimmune diseases. However, the author must understand that the strength of a such futuristic concept as a review article demands a lot of references, to state the observations made have some potential ground to explore in the future. Henceforth, the reviewer suggests authors few comments to make the study better;

- Include more references in each section, the reviewer noticed a pattern of       3-4 (approx) references per section only. 

- Expand section 5.2 (IBD) with more information and references.

Line 295: Why the numerical one is in superscript? 

Author Response

Response to Reviewer 3 Comments

First, the reviewer would like to appreciate the author's efforts to compile all this information to put forward the concept of altering the gut microbiome as a therapeutic strategy in autoimmune diseases. However, the author must understand that the strength of a such futuristic concept as a review article demands a lot of references, to state the observations made have some potential ground to explore in the future. Henceforth, the reviewer suggests authors few comments to make the study better;

Include more references in each section, the reviewer noticed a pattern of 3-4 (approx) references per section only.

Response 1: There have been additional references input in each section as well as a collective increase of 18 total new references to help increase the content of the paper.

Expand section 5.2 (IBD) with more information and references.

Response 2: Section 5.2 has been expanded to include more information as well as additional references.

“Currently, the pathogenesis of IBD is thought to include susceptible individuals that are exposed to gut antigens, such as microbiome changes which result in inappropriate activation of the immune system [52]. When experimental murine models of IBD are examined, under germ-free conditions, such as in the Britton et al. experiment, IBD is either not developed at all or significantly improved based on the expectation, which is highly suggestive of the role microbes play in IBD intestinal inflammation [33]. Further, alterations in microbiome components can be involved in different IBD subtypes. In a large study using 1815 stool samples from 303 patients with CD, 228 patients with UC, and 161 healthy controls, alpha diversity, meaning the observed taxa or evenness of a sample, was significantly lower in patients with both CD and UC compared with active controls, and active UC showed significantly lower alpha diversity than when stool was taken from the same subject during a time of inactive UC [53].”

Line 295: Why the numerical one is in superscript?

Response 3: The superscript has been changed with the correct in-text citation.

Reviewer 4 Report

The description of listed  AI diseases is partial/not complete, some informations from rheumatology are not exact. 70% of work is the description of diseases, 20% is related to microbiom and the discussion is extremely short. 

Author Response

Response to Reviewer 4 Comments

The description of listed AI diseases is partial/not complete, some informations from rheumatology are not exact. 70% of work is the description of diseases, 20% is related to microbiome and the discussion is extremely short.

Response 1: We appreciate the reviewer’s time and effort regarding our manuscript and their commentary. We feel that the description of the diseases was necessary to the context of the discussion regarding the microbiome. We have added additional content to the discussion and sections specific to the microbiome to help augment the content of those sections in relation to the disease description sections.

Round 2

Reviewer 2 Report

The justification that the authors have provided for the following:

'The concept of why gut microbiota is considered as regenerative medicine or an adjunct to it is not clear'- The underlying concept is clear but not completely validated.

Author Response

Response to Reviewer 2 Comments

The concept of why gut microbiota is considered as regenerative medicine or an adjunct to it is not clear'- The underlying concept is clear but not completely validated.

The gut microbiome is intimately related to human physiology as its mutualistic organisms perform many essential metabolic functions, including the synthesis of vitamins, amino acids, and cholesterol, and the fermentation of non-digestible substrates. Additionally, these microorganisms produce short chain fatty acids that regulate intestinal cell growth and differentiation and induce the secretion of mucin and other antimicrobial peptides, providing an essential protective role [46]. Alteration of the gut microbiome has the potential to be an effective adjunct therapy in Regenerative Medicine, supporting the body's own physiologic repair mechanisms and addressing the multifactorial causes of disease. By synthesizing antimicrobial compounds, outcompeting pathogenic organisms for nutrients and attachment sites, and directing signaling interactions with the adaptive immune system, clinical interventions to modify the gut microbiome hold promise as an adjunct and potentially primary therapy in Regenerative Medicine.

Reviewer 4 Report

I have checked the article until the line 262.  There are many  not correct medical informations and spelling mistakes. I  recommend the authors to consult a rheumatologist and ask for help to correct the article.  Please check and correct the whole article. 

Author Response

I have checked the article until the line 262.  There are many  not correct medical informations and spelling mistakes. I  recommend the authors to consult a rheumatologist and ask for help to correct the article.  Please check and correct the whole article. 

Response 2: We thank the reviewer for their suggestions. All the grammatical and spelling errors that were noted have been corrected. The manuscript has been edited appropriately, and all references used were rechecked to ensure that the infection presented in the manuscript is accurate.